



# Spatial distribution and occurrence probability of regional new particle formation events in eastern China

Xiaojing Shen[1], Junying Sun[1,2], Niku Kivekäs[3], Adam Kristensson[4], Xiaoye Zhang[1], Yangmei Zhang[1], Lu Zhang[1], Ruxia Fan[1], Xuefei Qi[1], Qianli Ma[5], Huaigang Zhou[6]

[1]State Key Laboratory of Severe Weather & Key Laboratory of Atmospheric Chemistry of CMA, Chinese Academy of Meteorological Sciences, 100081 Beijing, China

[2]State Key Laboratory of Cryosphere Science, Northwest Institute of Eco-Environment and Resources, Chinese Academy of Sciences, 730000 Lanzhou, China

[3] Finnish Meteorological Institute, P.O. Box 503, 00101 Helsinki, Finland

[4] Division of Nuclear Physics, Lund University, P.O. Box 118, SE-221 00 Lund, Sweden

[5]Lin'an Atmospheric Background Station, Meteorological Bureau of Zhejiang Province, Hangzhou 311300, China

[6]Environmental Meteorology Forecast Center of Beijing-Tianjin-Hebei, Beijing 100089, China

*Correspondence to* J. Y. Sun (jysun@camscma.cn), N. Kivekäs (Niku.Kivekas@fmi.fi) and A. Kristensson (adam.kristensson@nuclear.lu.se)

**Abstract**. In this work, the spatial extent of new particle formation (NPF) events and the relative probability of observing particles originating from different spatial origins around three rural sites in eastern China were investigated with the NanoMap method, using particle number size distribution (PNSD) data and air mass back trajectories. The length of the datasets used were 7-year, 1.5-year and 3-year at rural sites Shangdianzi (SDZ) in North China Plain (NCP), Mt. Tai (TS) in central eastern China, and Lin'an (LAN) in Yangtze River Delta region in eastern China, respectively. Regional NPF events were observed to occur with the horizontal extent larger than 500 km at SDZ and TS, favored by the fast transport of northwesterly air masses. At LAN, however, the spatial footprint of NPF events was mostly observed around the site within 100-200 km. Difference in the horizontal spatial distribution of new particle source areas at different sites was connected to typical meteorological conditions at the sites. Consecutive large-scale regional NPF events were observed at SDZ and TS simultaneously and were associated with a high surface pressure system dominating over this area. Simultaneous NPF events at SDZ and LAN were seldom observed. At SDZ the polluted air masses arriving over NCP were associated with higher particle growth rates (*GR*) and new particle formation rates (*J*) than air masses from Inner Mongolia (IM). At TS the same phenomenon was observed for *J*, but *GR* was somewhat lower in air masses arriving over NCP compared to those arriving from IM. The capability of NanoMap to capture the NPF occurrence probability depends on the length of the dataset of PNSD measurement, but also on topography around the measurement site and typical air mass advection speed during NPF events. Thus the long-term measurements of PNSD in planetary boundary layer are necessary in the further study on spatial extent and probability of NPF events. The spatial extent, relative probability of occurrence and typical evolution of PNSD during NPF event presented in this study provide valuable information to further understand the climate and air quality effect of new particle formation.



## 1. Introduction

Atmospheric new particle formation (NPF) from gaseous precursors is a major source of particles in the atmosphere in many regions worldwide (Kulmala et al., 2013). Due to the growth by condensation and coagulation processes, these particles can reach sizes where they act as cloud condensation nuclei (CCN), thereby affecting cloud formation and climate

(Kerminen et al., 2012), and contribute to the particle mass enhancement even in polluted regions (Guo et al., 2014).

The continental regional NPF events can occur over a large spatial scale (Kulmala et al., 2004). The spatial scale of NPF events or horizontal extent of the air mass in which NPF events occurs can help to identify the possible particle sources. The large variability in spatial distribution of regional NPF does, however, cause large uncertainty in modeling of the place and frequency of NPF events (Hussein et al., 2009). Furthermore, the properties of NPF events as observed at a stationary

measurement site are also affected by processes and conditions taking place hundreds of kilometers upwind of the site (Kivekäs et al., 2016).

There have been a number of studies addressing the spatial scale of NPF events (e.g. Birmili et al., 2003; Komppula et al., 2006; Charron et al., 2007; Hussein et al., 2009; Németh and Salma, 2014; Väänänen et al., 2016) where the horizontal scale of NPF has been found to be from hundreds to thousands of kilometres. Many of these studies have used one or several

measurement sites as the bases of their analysis. Some of them were conducted based on stationary measurements at more than one ground-level stations simultaneously (Hussein et al., 2009; Németh and Salma, 2014; Komppula et al., 2006) and some of them use sites at different altitudes in the same area (Boulon et al., 2011; Birmili et al., 2003). There have also been airborne measurements on the spatial scale of NPF events (Väänänen et al., 2016). The flight measurements are efficient in deriving the vertical and horizontal extents and characteristics of a NPF event, but are technically demanding and costly.

Hussein et al (2009) showed that the longer the time span when the newly formed particle mode can be followed at a stationary measurement site is, the larger the horizontal extent of that NPF event would be upwind of the site. Kristensson et al. (2014) processed this idea further and developed the NanoMap method for estimating the spatial distribution of regional NPF events based on continuous particle number size distribution (PNSD) measurement at a single point and back trajectory analysis.

In China, where the particle pollution is a serious issue, there have in the recent years been several studies addressing NPF events based on long-term field measurements (Shen et al., 2011; 2016a; 2016b; Wu et al., 2007; Herrmann et al., 2014; Kivekäs et al., 2009). They have discovered that NPF takes place frequently, ~30-40 % of the days. It has also been reported that NPF events observed in Beijing can sometimes extend over one hundred kilometers (Wang et al., 2013). But the study of spatial distribution of the NPF events based on long-term dataset in China has not been reported yet.

In this study, we apply NanoMap method (Kristensson et al., 2014), using back trajectories and PNSD measurements at




three regional background stations in China: Shangdianzi (SDZ) in North China Plain (NCP), Mt. Tai (TS) in central eastern China (CEC), and Lin'an (LAN) in Yangtze River Delta (YRD) region in eastern China. We aim to detect the areas upwind of the sites where the observed new particles have been formed (horizontal extent of each event) and the areas where the new particle formation typically takes place. Finally, the PNSD characteristics, new particle formation rate ($J$), particle growth

rate ($GR$) and pre-existing condensation sink ($CS$) are quantified during NPF events at these stations depending on the origin of the air masses. This analysis identifies the meteorological and air mass characteristics and geographical source areas responsible for the observed NPF events and the spatial scale of NPF, which can be used to quantify the regional impact on the aerosol population and climate through CCN production, as well as the air quality.

## 2    Experiments and method

### 2.1  Station location

The SDZ station (40°39'N, 117°07'E, elevation of 293 m asl, above sea level) (Fig. 1) is one of the regional stations of Global Atmosphere Watch (GAW) in China. This station is located in the northern part of the NCP region, in the Miyun District of Beijing, ~150 km northeast of the urban area of Beijing and 55 km northeast of the county of Miyun District with a population of ~0.5 million. The sampling site is situated on the south slope of a hill, in a valley with a northeast-southwest

orientation. SDZ is frequently influenced by air masses passing over the heavily polluted regions of Beijing, Tianjin and Hebei Provinces (Shen et al., 2011). Beijing and Tianjin are main megacities in NCP region with populations over 20 million and 15 million, respectively, with a high population density. Hebei province has a population of over 70 million, with the economy dominated by agriculture and industry. Generally speaking, Beijing-Tianjin-Hebei is the dominant particle emission source in the NCP region.

The measurement site at Mt. Tai (TS, 40°15'N, 117°06'E, elevation of 1534 m asl) (Fig. 1) is positioned at the Taishan National Basic Meteorological Observation Station at the top of Mt. Tai in Shandong Province. Mt. Tai is the highest peak in its surroundings within a 300 km radius. It is a fault-block mountain, higher in the south than north and about 80% area covered by vegetation. The station receives some local anthropogenic emissions from tourism on the mountain and from the city of Tai'an with a population of ~ 5.5 million at the foothill to the south of Mt. Tai (Shen et al., 2016a) and Ji'nan with a population of

~7.1 million located about 60 km north of the site. The entire Shandong province around Mt. Tai is densely populated with a total population of ~100 million.

The LAN (30°17'N, 119°45'E, elevation of 139 m asl) station is also one of the regional GAW stations in China, located in the YRD region (Shen et al., 2016b). YRD, including the provinces of Anhui, Jiangsu and Zhejiang, as well as Shanghai city, is one of the most economically active regions in China. LAN station is located ~50 km west of Hangzhou with a



population of ~9 million, and ~200 km southwest of Shanghai, a megacity with a population of ~20 million.

### 2.2 Instrumentation

The PNSD measurements at all three sites were performed with Twin Differential Mobility Particle Sizer (TDMPS) systems, built by TROPOS, Germany, consisting of a Differential Mobility Analyzer (DMA) of Vienna-type median length

(Winklmayr et al., 1991) and an Ultrafine DMA of Vienna-type short length (Winklmayr et al., 1991) in parallel to size-classify the sampled aerosol particles. These are followed by a Condensation Particle Counter (CPC 3772, TSI Incorporated, USA) and an Ultrafine CPC (CPC 3776, TSI Incorporated, USA), respectively, to count the number of particles after each DMA. The TDMPS systems measured in the size range (3-900 nm) every 10 min. All ambient aerosols were dried with an automatically regenerating Silica gel absorption dryer system developed by TROPOS, Germany, to keep

the relative humidity of the air sample below 40% (Tuch et al., 2009). All diffusion losses in the dryer and tubes were corrected (Wiedensohler et al., 2012) and the raw mobility distribution data was converted to PNSD using the inversion routine described by Pfeifer et al. (2014). Finally, the particle number concentration was corrected to the standard atmosphere conditions (1013.25 hPa, 273.15K). At TS station, TDMPS system was operated since December 2010. Before that (July to November 2010), the PNSD for mobility diameter was measured by a Scanning Mobility Particle Sizer (SMPS,

Model 3080, TSI Incorporated, USA) in size range of 10-680 nm.

The same TDMPS instrument was used at TS and LAN, excluding the possibility of simultaneous measurements at these two sites. Another TDMPS was operating at SDZ during the entire measurement period. The duration of the datasets used for analysis in this study were 7 years (1 March, 2008 to 31 December, 2014) at SDZ, 1.5 years (1 July, 2010 to 31 December, 2011) at TS and 3 years (1 January, 2013 to 31 December, 2015) at LAN.

### 2.3 NanoMap

NanoMap is a method which can give an estimation of where new particles are formed up to 500 km distance upwind of any field station during an NPF event observed at that station (Kristensson et al., 2014). It is based on PNSD measurement data and back trajectories and has been validated using the Finnish field site Hyytiälä as test station (Kristensson et al., 2014), and has further been used at a European continental site, Budapest, Hungary (Nemeth and Salma, 2014). The NanoMap

method follows four steps: 1) Classification of NPF events and identification of class 1 events (Dal Maso et al., 2005), 2) determining the start and end time of the particle formation at the lowest size bin of the measured PNSD, 3) estimating the time when the newly formed particle mode can no longer be identified (end of growth time=EOG time), and 4) plotting a geographical map with the spatial distribution of NPF events based on the above defined time points and air mass back trajectories. In NanoMap, two assumptions need to be met: 1) the formation of particles is assumed to take place at the same



time in a large area, and 2) the particle formation is assumed to take place also at the measurement site, even though those particles are too small to be detected. NanoMap is further affected by subjectivity, as the user determines the starting ($t_s$) and ending times ($t_e$) of the NPF event, as well as the end of growth time ($t_{eog}$) based on his / her own judging from the measurement data.

NanoMap can also be used for calculating a relative NPF occurrence probability by dividing the number of times NPF event was detected in a given grid cell by the total number of trajectories passing through that grid cell. While the uncertainty of the probability with this method is rather small in grid cells with a high number of passing trajectories, it can be relatively large in grid cells with a low number of passing trajectories, typically located further away from the measurement site. In this study, the grid resolution was chosen to be 0.2°×0.4° latitude-longitude.

In this study the lowest detection limit diameter of the TDMPS systems were 3 nm at all three sites and that of the SMPS was 10 nm at TS during the time period specified above in chapter 2.2. The nucleated particles need to grow from the initial size of 1.5 nm (Kulmala et al., 2013) to the detection limit, which takes some time. Thus, we determined the time-shift between the real formation time and observed formation time (Kristensson et al., 2014). This was done based on the reported mean $GR$ at each site (Shen et al., 2016b), calculated from the entire event from the start time to the EOG time. The mean

growth rates were 3.6, 6.0 and 6.2 nm h$^{-1}$ at SDZ, TS and LAN, respectively, leading to time-shift of 0.42 h, 0.25 h and 0.24 h at SDZ, TS and LAN, respectively. From July to November 2010 at TS, the time-shift was be 1.42 h as the detection limit diameter was 10 nm.

## 2.4  Classification of NPF events

The parameters characterizing NPF events, including $J$, $GR$ and $CS$, can be calculated based on the PNSD measurement

(Dal Maso et al., 2005; Kulmala et al., 2001). In this work, $J_{3-10}$ included the net flux of particle number concentration into 3 nm (the lowest detection limit of TDMPS system) to 10 nm size range and the loss due to coagulation process in this size range. For the SMPS data from TS, $J_{10-25}$ was calculated with the particle in the size range of 10-25 nm. $GR$ was calculated from the evolution of the representative geometric mean diameter (GMD) of the entire measured PNSD at each time step, $D_p$, from $t_{start}$ to the EOG time. In this study, $D_p$ is derived by a fitting algorithm of PNSD based on two lognormal modes

(Hussein et al., 2009). The $CS$ is calculated based on the assumption that the properties of condensable vapors were similar to sulfuric acid, which has been shown to be a major component participating in the nucleation process (Kulmala et al., 2013), and is also abundant in eastern parts of China (Qi et al., 2015; Wang et al., 2011).

## 2.5  Meteorological back trajectories

The transport of air masses used for NanoMap analysis was calculated with the HYSPLT model (Draxler and Hess, 1998).



In this study, we calculated 72-h back trajectories with 1-h resolution at each station. The input meteorological data (Global Data Analysis System data, GDAS) in the HYSPLIT model is used with one-degree latitude-longitude resolution. In the GDAS data the terrain height is 380 m asl at SDZ, 150 m asl at TS and 170 m asl at LAN, respectively. In case of TS, this terrain height is clearly not representative, as the actual site altitude is 1540 m asl. For the other two sites the model terrain

height is roughly correct. The trajectory ending heights at the three stations were chosen as 500 m agl (above ground level) at SDZ, 1500 m agl at TS and 500 m agl at LAN station, respectively, being slightly above the real site altitudes of all three sites.

## 3   Results and Discussion

### 3.1   Frequency of NPF event at the three sites

The NPF characteristics and frequencies at all three sites have been published by Shen et al., (2016b) based on the datasets including March 2008-December 2013 at SDZ, January-December 2011 at TS and January-December 2013 at LAN, respectively. The different length of datasets used in this study compared to Shen et al. (2016b) resulted in a slight difference in the statistics of NPF events due to the inter-annual variation of NPF occurrence and characteristics. The number and fractions of different NPF event types and non-NPF days as used in this study at three sites are given in Table 1. At SDZ site,

there were in total 1970 measurement days from 2008 to 2014, among which 1525 days were used in the analysis while the others were considered as invalid days due to instrument malfunction, calibration, etc. The statistics of NPF event occurrence at TS was based on only 1.5-year of measurements, showing an NPF event frequency of 32% of all days. There was a 3-year dataset used for the analysis at the LAN station, demonstrating a lower NPF fraction compared with SDZ and TS. The NPF occurrence frequency also showed an inter-annual variation, with lowest value of 16% in 2015 and the highest

value of 28% in 2013. In the previous study at LAN, it has been reported that $CS$ on NPF days was 0.03 s$^{-1}$, higher than that at SDZ and TS, ~0.02 s$^{-1}$ (Shen et al., 2016b). This indicated higher concentration of pre-existing particles, which is not a favorable factor for NPF occurrence at LAN.

### 3.2   Where does the new particle formation take place?

The back trajectories of air advecting from the potential places where the observed new particles were formed at 1.5 nm

were calculated and analyzed with the NanoMap method for each station (Fig. 2). The general conclusion is that at SDZ and TS sites, northwesterly air masses were dominating during the NPF events. The result is consistent with the previous studies reporting that the northwesterly air masses were connected with clear, dry and clean conditions, favoring new particle formation (Shen et al., 2011). Under this condition, the northern China is usually governed by the northern continental



high-pressure system or a cold frontal passage. The further extent of NPF events occurring in northwesterly direction from

SDZ and TS were caused by a higher wind speed related to these synoptic systems. SDZ and TS are located about 500 km

apart, and are often under the same synoptic weather system.

But for the LAN station, which is located about 1200 km from SDZ and 700 km from TS, the air masses connected to

NPF arrived from more diverse wind directions with generally lower advection speeds. The relatively low wind speed is

related to the weather conditions influenced by continental and Pacific high-pressure systems. Quite often, it is usually to

follow the growth of a nucleation mode during NPF within 10-15 hours. With a frequency of ~80% of a wind speed ~3 m s$^{-1}$

or lower at LAN, this means that the maximum horizontal extension of the observed NPF event according to NanoMap is

restricted to within about 200 km. However, NPF event might take place beyond these 200 km as well, but it cannot be

verified with NanoMap for the conditions typical to LAN.

The topography around TS is complex, consisting of continuous hills and mountains with peak heights lower than 1000 m

asl, extending over a large area of about 400 km$^2$. The limited vertical resolution and imperfect topography information in

the HYSPLIT model leads to a higher uncertainty of back trajectories than at areas with more simple topography. Therefore,

we examined the sensitivity of the NanoMap results at TS with two additional trajectory ending heights, 1000 and 2000 m

agl. The different elevations however, didn't have a significant influence on the NanoMap results, indicating that the air was

well mixed during the NPF events. For this reason, this issue was not pursued further.

The vertical profiles of the calculated 72-h back trajectories arriving at SDZ, TS and LAN with the terminating height of

500, 1500 and 500 m agl, respectively, were also analyzed (Fig. 3). At SDZ and LAN majority of the back trajectories

originated from an altitude below 2000 m agl and travelled at a lower altitude around 1000 m agl when within 500 km radius

from the measurement sites, indicating strong interactions of the air and ground surface within the planetary boundary layer

(PBL). At TS, however, majority of the trajectories travelled at an altitude above 1500 m agl, representing free troposphere

(FT) air with only a limited influence of PBL, indicating the presence of NPF also in the free troposphere.

### 3.3  NPF occurrence probability

Fig. 4a, c, e show how many times the formation of new particles was observed to take place in each grid cell around the

stations. There were more NPF-connected trajectories passing over each grid cell in the SDZ data than in the data for the

other sites since the SDZ dataset was much longer. We also estimated the relative probability of new particles being formed

in each grid cells around each measurement site (Fig. 4b, d, f). According to Kristensson et al. (2014) at least three years

dataset is needed for a reliable NPF probability prediction with NanoMap, so we can be quite confident for the NPF

probability results for the SDZ data. At SDZ, the probability distribution in Fig. 4b was quite consistent with that of back

trajectory count in Fig. 4a with some exceptions. For example, the northerly air masses are more pronounced in the





probability data than in the number of observed cases of NPF (Fig. 4a, b). The southerly air masses advecting from the nearby areas were also associated with high probability of NPF, demonstrating that regional NPF events in NCP region could also occur under highly polluted conditions.

For the TS site, the shorter dataset of 1.5-year was not enough for a robust prediction of the NPF probability. The NPF probability analysis showed high probability in many grid cells where the number of trajectory counts was low (Fig. 4c, d). In addition, the high altitude of TS in respect to its surroundings creates further uncertainties. The shifts between PBL and FT air can alter both the particle population characteristics and the source of the air observed.

For the LAN station, when the air mass originated from the area more than ~200 km away, the NPF probability had somewhat larger uncertainty due to less trajectory passages in these grid cells. At LAN, the air mass moving speed was typically lower than at the other sites, leading to air travelling only about 200 km within 72 hours most of the times. This indicates that the capability of NanoMap to predict probability of NPF around LAN is spatially limited due to the slow advection of air, despite that we actually have a 3-year dataset at LAN. The above discussion reveals that both the length of the dataset and the wind speed govern the applicability of the NPF probability prediction in the NanoMap analysis. Nevertheless, from Fig. 4f it is possible to see that no air mass arriving direction is pronounced in the NPF probability around LAN.

### 3.4  Large scale regional NPF events

Based on the measurements conducted simultaneously at several stationary sites, we could also identify the spatial scale of regional NPF events with a different method than NanoMap (Crippa and Pryor, 2013). During the period of simultaneous measurements at SDZ and TS in the second half of 2010 and 2011, there were 54 simultaneous NPF events at the two sites, accounting for approximately 50% to the total NPF events at both SDZ and TS during this period. There were, however, less than 10% of the regional NPF events observed simultaneously at SDZ and LAN in 2013 and 2014, as was expected. These two stations are almost 1200 km apart, making it difficult for them to be affected by the same homogenous air mass. Unfortunately, there were no simultaneous PNSD measurements at TS and LAN. Previous studies have reported that regional NPF events were observed over large areas (Hussein et al., 2009) and favored by the air masses corresponding to cold air advection (Charron et al., 2007; Sogacheva et al., 2005). High-pressure systems resulted in more NPF events at SDZ and TS, with the north wind field dominating. This synoptic pattern was favorable for turbulent mixing and dilution of particle-rich boundary layer air with cleaner air above, which resulted in lower *CS* favoring regional NPF event. This finding is in line with another study in Egbert, Ontario, Canada (44°14'N, 79°47'W, 251m asl) (Pierce et al., 2014). However, the class of NPF events without a clear growth process and some undefined cases of nucleation burst were also observed



simultaneously at both SDZ and TS. These were probably influenced by the local conditions, such as the meteorological

factors, precursors and pre-existing particles.

Several consecutive regional NPF events occurred simultaneously at SDZ and TS on 18-22 September 2011 (Fig. 5). The

two stations are located about 500 km apart, indicating that the regional NPF event could occur across such a large spatial

scale. The cases at the two locations were similar in the start time and temporal behavior, even though showing different

absolute values of $J$ and $GR$. The analysis of meteorological conditions indicated that the incursion of cold air masses

associated with a stagnant high pressure system could lead to stronger winds and probably strong turbulent mixing, favoring

the occurrence of NPF events. The back trajectories from north and northwest typically bring clean and dry air masses often

connected to NPF at SDZ and TS (Shen et al., 2011; Shen et al., 2016a). The synoptic charts of mean sea-level atmospheric

pressure during this case were taken from the US National Center for Environmental Prediction (NCEP) reanalysis dataset.

The data were available every 6 h (00, 06, 12 and 18 UTC) with 2.5°×2.5° resolution (Fig. 6a). The 72-h back trajectory

results at SDZ and TS are shown in Fig. 6b.

### 3.5  Typical PNSD and NPF parameters under different NPF-connected air masses

The geographical map was classified into four sectors according to the main direction of the back trajectories, which can

also reflect the different air mass origins influencing each station on NPF event days. These major sectors at all three sites

are defined as: north (covering the direction of 0-45° and 315-360°), east (45-135°), south (135-225°) and west (225-315°),

respectively. Due to the different locations of the sites some potential particle source areas are located in different sectors in

respect to different measurement sites. The back trajectories arriving from south at SDZ and from north at TS both represent

the air masses arriving over NCP region. The westerly air masses arriving at SDZ and TS are both mainly from Inner

Mongolia (IM), but at TS they have traveled over parts of NCP region. The air masses from IM and NCP region were the

dominant air masses at SDZ and TS during NPF events, based on the NanoMap result as discussed above. At LAN, the air

masses from all directions were connected to NPF according to the trajectory count and probability analysis in Fig. 4e-f.

The evolution of PNSD during different periods of the observed NPF events at each station influenced by different

NPF-connected air masses was given in Fig. 7 as the hourly mean PNSD of the pre-existing particles (2 h before NPF event

started, 2 h pre-NPF), 5 h and 10 h after the NPF event started (5 h aft-NPF and 10 h aft-NPF) and at EOG time was

calculated. At SDZ the PNSD showed higher number concentration and larger GMD in all stages of the NPF event air

masses arriving from NCP than in the air masses from IM. However, at TS the IM and NCP air masses resulted in similar

GMD of PNSDs, but higher number concentration when NCP regional air mass dominated. The faster increase of GMD of

particles in IM air observed at TS (compared to IM air at SDZ) is probably resulted by the high concentration of condensable



vapors accumulated in the air as it passes over parts of NCP before arriving at TS. The GMD at TS did not increase between

10 h after the NPF event started and EOG time, indicating that the growth process usually stopped within or near 10 h from

the beginning of the NPF event. This might be due to the fact that the TS station is influenced by the local mountain-valley

breeze and the transition between PBL and FT due to its high elevation (Shen et al., 2016a). At SDZ and LAN, the EOG

time was usually longer than 10 h, which can be seen in Fig. 7d, f, as increase of the GMD between 10 h during NPF event

and EOG.

The statistics of $J_{3-10}$, $GR$ and $CS$ in NPF events influenced by different air masses at each station are given in Table 2. It

shows that at SDZ $J_{3-10}$ and $GR$ are about twice as high for particles originating from NCP region compared to particles from

Inner Mongolia. In NCP region, the air mass passed over polluted areas, favoring more vapors to condense on the newly

formed particles under the growth process. It could thus result in the high $CS$, which was four times higher in the NCP

regional air masses than in the IM air masses. At TS, the NCP regional air masses showed a higher $J_{3-10}$ but a similar or

slightly lower $GR$ as compared with the IM, and also a similar $CS$. The difference between the $GR$ behavior between SDZ

and TS could be affected by the complexity of the air masses arriving at TS as discussed above or the less robust statistics at

TS due to smaller amount of data. It should also be noted that the IM air arriving at TS had passed some parts of NCP region

and were more polluted than the IM air mass at SDZ. This resulted in the significantly higher $GR$ and $CS$ in IM air mass at

TS than the same air mass at SDZ.

## 4. Conclusions

In this work, we applied the NanoMap method on the particle number size distribution (PNSD) measurements at three

locations under rural environments in eastern China. With NanoMap, we evaluated the upwind locations where the particle

formation took place on the NPF event days observed at the sites and the relative probability of observable NPF event to take

place around the sites was also calculated. We focused on the regional NPF events, in which the nucleated particles could

grow to the size to be the potential CCN. The statistical results based on the 7-year, 1.5-year and 3-year measurements at the

respective stations of SDZ, TS and LAN showed this type of NPF events to account for about 15-29% to the total

measurement days depending on the station. During NPF event days, the most common air masses were northwesterly at

both SDZ and TS. The observable formation of new particles could extend up to about 700 km to the northwest and were

quite similar at SDZ and TS, which are located about 500 km apart. The occurrence of NPF events from northwest are many

times governed by a homogenous high-pressure system. But at the LAN station, the horizontal extent of observable NPF was

constrained to about 200 km from the station during NPF events, independently of the air mass arriving direction. This

difference in the horizontal extent of observable NPF event between SDZ, TS and LAN was probably caused by the synoptic





conditions. It was also found that at the low elevation sites, SDZ and LAN, the NPF-connected air masses approaching the sites usually travelled in the planetary boundary layer (PBL). For TS, however, the majority of NPF-connected air masses originated from free troposphere (FT). The shifts between PBL and FT can cause limitations to the applicability of NanoMap at mountain sites such as TS.

The prediction of NPF occurrence probability was more reliable at SDZ due to the long dataset there. The data at SDZ showed that high NPF probability was related to the northwesterly (Inner Mongolia) and southerly (NCP regional) air masses. This coincided with the high number of NPF events in these wind directions. The prediction of NPF probability around TS and LAN was less robust due to shorter datasets, complex topography at TS, and the low travelling speed of air masses arriving at LAN.

Regional NPF events occurred simultaneously at SDZ and TS when the two sites were influenced by the same surface high-pressure systems. The representative mean PNSDs during different phases of the NPF events under different air masses are also given for each site. This analysis revealed that the NCP regional air mass resulted in higher particle number concentration and larger geometric mean particle diameter at SDZ, as well as higher formation rate and growth rate than Inner Mongolia air masses. This could be explained by higher concentrations of low volatile precursors participating in the

nucleation and growth processes when the air mass passed through more polluted areas. The Inner Mongolia and NCP air masses showed somewhat different impacts on the nucleation and growth process on NPF events between SDZ and TS, which was probably influenced by the fact that IM air cannot reach TS without passing over some parts of NCP, and also by the high elevation of TS site. EOG time at TS was also often shorter than 10 hours, but it was longer at SDZ and LAN, which also has an effect on the analysis.

Our results can help to better understand the aerosol sources in these regions and could also improve the air quality modeling and regional climate modeling work. This work also addressed the uncertainty of the NanoMap methods in terms of dataset length and spatial extent of reliable results. The uncertainty of NPF occurrence probability determined by NanoMap is tightly linked with the length of dataset and the spatial extent of reliable results is further linked to the advection speed of air mass during NPF events. In addition, the topography of TS resulted in further uncertainty. In general, it is clear

that more long-term PNSD measurements are necessary in the future to study and evaluate the climate effect of NPF events. Furthermore, simultaneous PNSD measurements at several sites in the same region are needed to reliably estimate the horizontal extent of NPF events.

*Acknowledgement*. This work is supported by National Key Research Program of China (grant no. 2016YFC0203306), key

project of CAMS (grant no. 2017Z011, 2016Z001, 2016Y004), the CMA Innovation Team for Haze-fog Observation and





Forecasts and the Swedish FORMAS research grant (project no. 2010-850). The work is also supported by the European Commission I3-project ACTRIS-2 (grant no. 654109) and Academy of Finland Center of Excellence program (grant no. 307331). This work contributes also to the PEEX-initiative. The authors would also like to thank Prof. Tareq Hussein for making his particle mode-fitting code available for this work.

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

**Table 1. The number count and frequency (%, in brackets) of different types of NPF event days and non-event days observed at**
**SDZ, TS and LAN, respectively, during the analyzed measurement days.**

| Station | Class 1 | Class 2 | Total NPF | Undefined | Non-NPF |
|---------|---------|---------|-----------|-----------|---------|
| SDZ | 442 (29) | 136 (9) | 578 (38) | 283 (19) | 664 (43) |
| TS | 97 (24) | 32 (8) | 129 (32) | 105 (26) | 172 (42) |
| LAN | 145 (15) | 64 (7) | 209 (22) | 126 (13) | 611 (65) |



**Table 2. The mean and standard deviation of formation rate ($J_{3\text{-}10}$), growth rate ($GR$) and condensation sink ($CS$) of NPF influenced by different air mass at each station, respectively.**

| Station | Air mass | $J_{3\text{-}10}$ (cm$^{-3}$ s$^{-1}$) | $GR$ (nm h$^{-1}$) | $CS$ (s$^{-1}$) |
|---------|----------|------------------|-----------|---------|
| SDZ | Inner Mongolia | 5.1±3.1 | 3.3±1.6 | 0.01±0.01 |
|  | NCP regional | 10.2±6.5 | 6.3±2.9 | 0.04±0.03 |
| TS | Inner Mongolia | 2.9±1.5 | 5.8±3.2 | 0.02±0.01 |
|  | NCP regional | 4.1±2.1 | 4.5±2.0 | 0.02±0.01 |
| LAN | YRD regional | 4.6±3.4 | 7.4±4.3 | 0.04±0.02 |

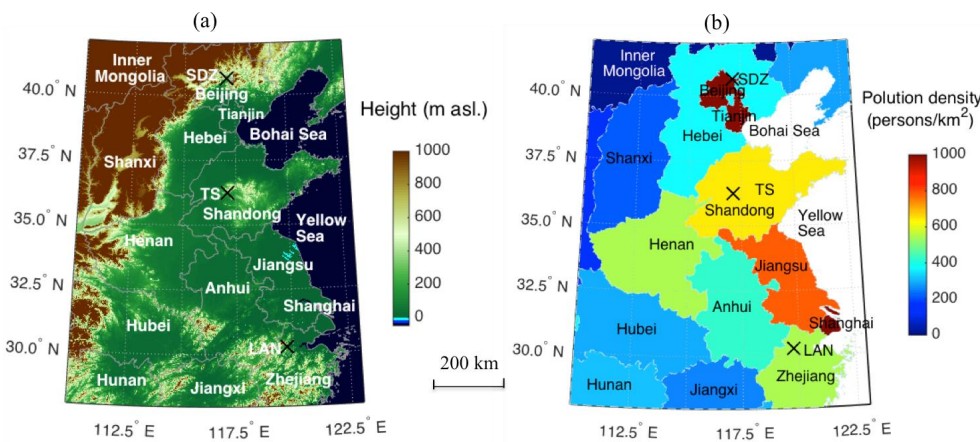

**Fig. 1. The location of Shangdianzi (SDZ), Mt. Tai (TS) and Lin'an (LAN) sites. The colors indicate the (a) terrain height above sea level (http://ngcc.sbsm.gov.cn) and (b) population density in different provinces based on the statistics at the end of 2014 (http://data.stats.gov.cn/).**





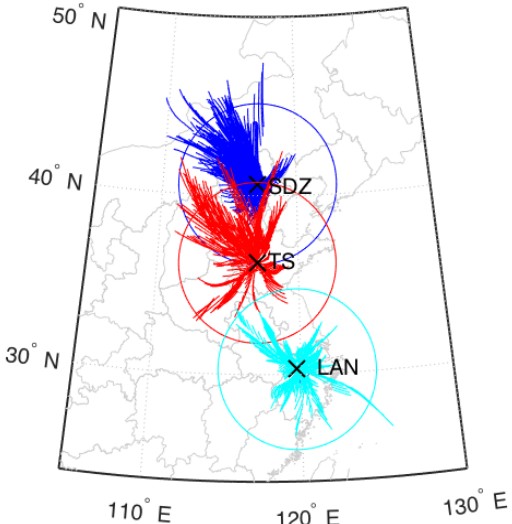

**Fig. 2. 72-h back trajectories arriving at SDZ (blue), TS (red) and LAN (cyan) from the areas where new particles were observed to be formed according to NanoMap. The stations are marked as crosses, and circles indicate the horizontal radius of 500 km from the station.**

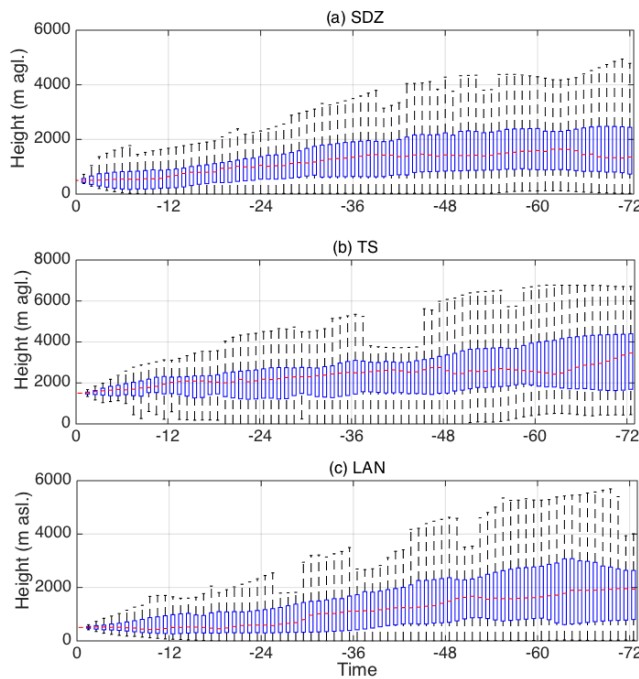

**Fig. 3. The travelling height (above ground level) of the 72-h back trajectories arriving at SDZ, TS and LAN, respectively. The red lines represent the median height, the edges of the blue boxes are 25th and 75th percentiles and the upper and lower edges of black dot lines are the 95th and 5th percentiles.**



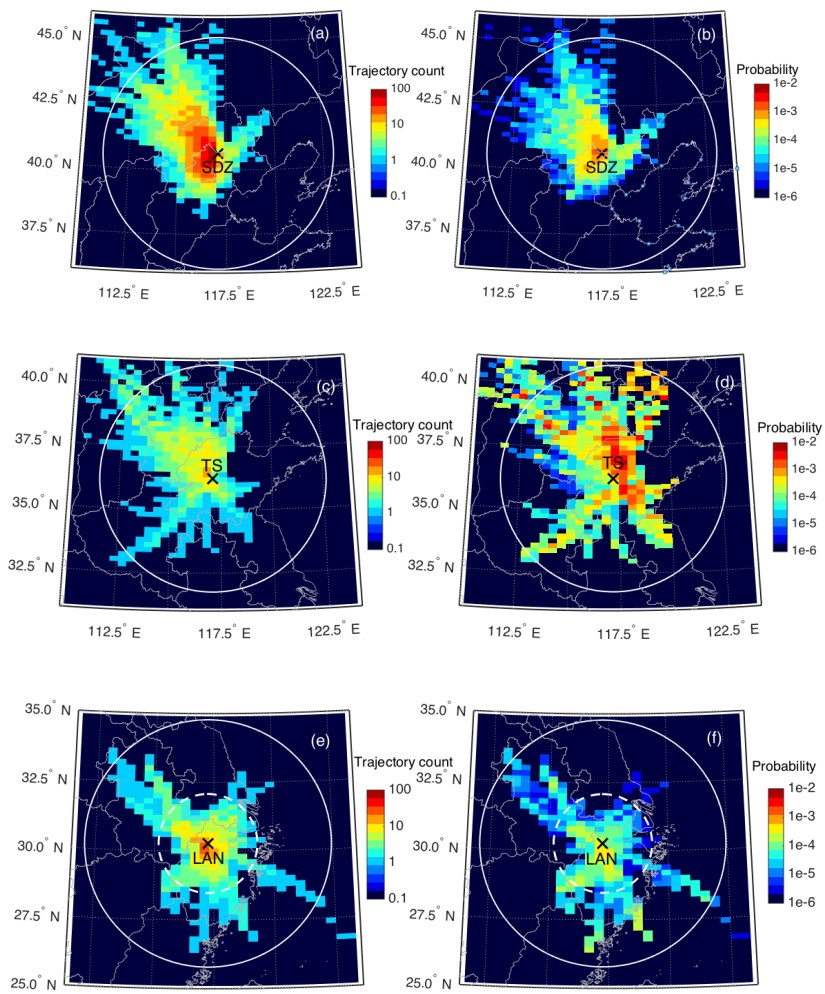

**Fig. 4. The number of trajectory counts in each grid cell for new particle formation during NPF events (left panel) and relative probability of NPF occurrence (right panel) at SDZ (a, b), TS (c, d) and LAN (e, f) stations, respectively. The stations are marked with crosses, and the solid and dashed white circles indicate the horizontal radius of 500 km and 200 km from the station, respectively. The grid resolution is 0.2°×0.4° latitude-longitude.**



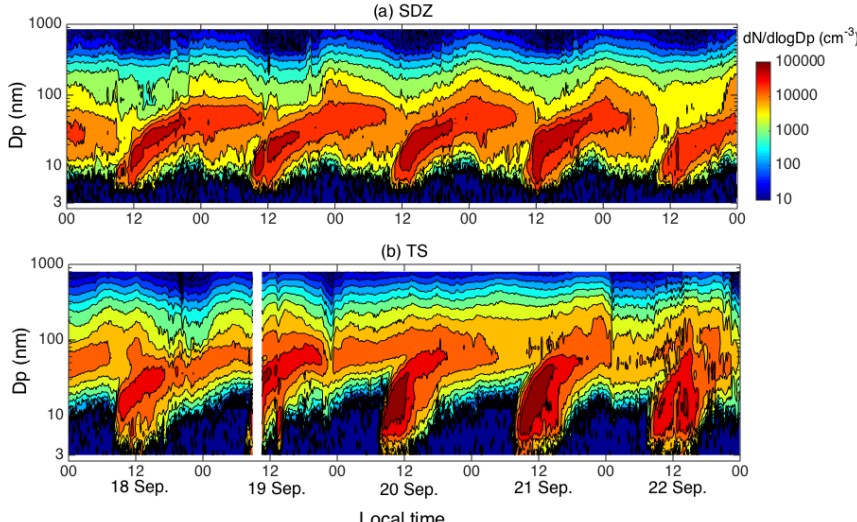

**Fig. 5. Five consecutive regional NPF events occurred at SDZ (a) and TS (b) simultaneously on 18 to 22 September 2011.**

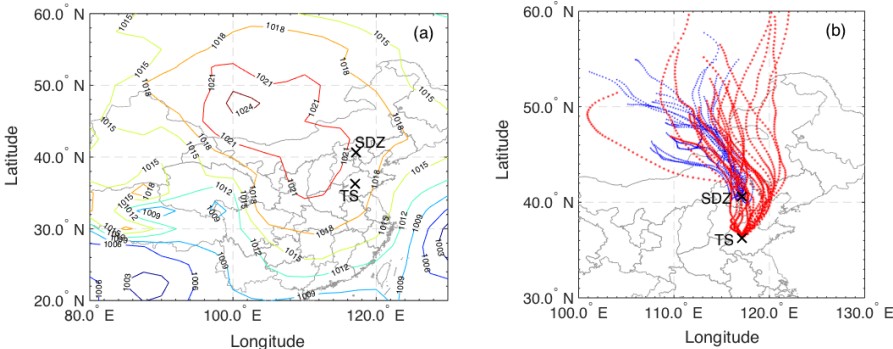

**Fig. 6. Mean atmospheric pressure (hPa) at sea-level represented by colorful lines (a) and 72-h back trajectories arriving at SDZ**
5   **(blue lines) and TS (red lines) (b) on 18 to 22 September 2011, at four times per day (00, 06, 12, 18 UTC).**





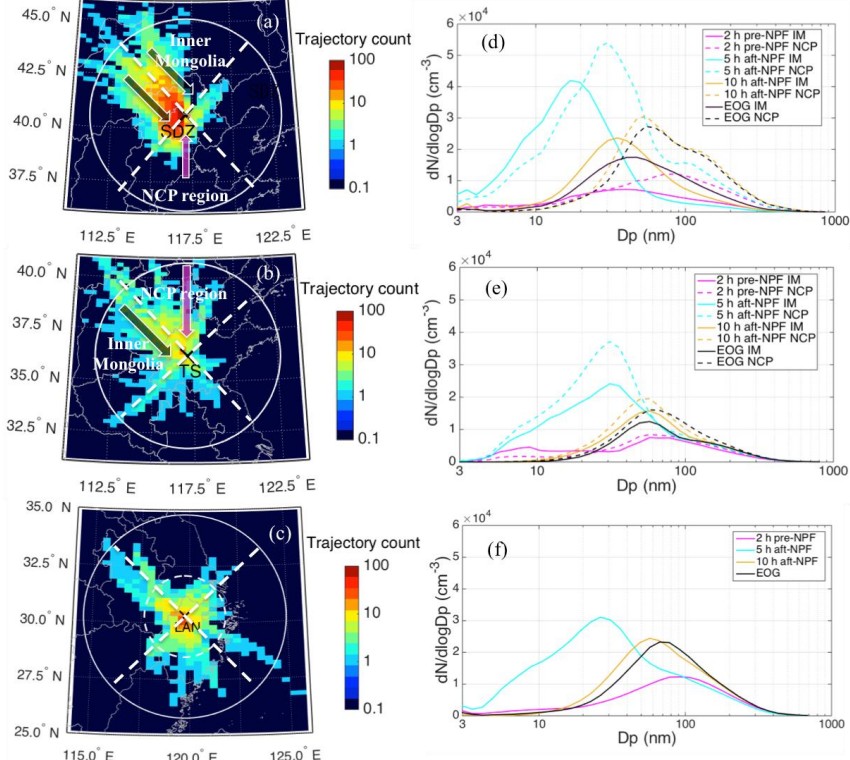

**Fig. 7. Classification of the observed source regions of new particle formation (left panel), and mean PNSD (right panel) at 2 h before NPF started (2 h pre-NPF), 5h after NPF started (5 h aft-NPF), 10h after NPF started (10 h aft-NPF) and EOG time corresponding to air mass from Inner Mongolia (IM) and NCP region (NCP) at SDZ (a, b), TS (c, d) and LAN (e, f), respectively.**

5     **The solid and dashed white circles in left panel indicate the horizontal radius of 500 km and 200 km from the stations, respectively.**