# Peer review of "Spatial distribution and occurrence probability of regional new particle formation events in eastern China"

_Atmospheric Chemistry and Physics, 2017_

## Referee Comment (RC1) · Anonymous Referee #1 · 29 Sep 2017

The manuscript deals with the spatial extent of new particle formation based on long-term particle number size distribution (PNSD) measurements in China. The work presented here describes the NPF occurence probability and the possible origin of the nucleating air masses as well. A case study is also performed. The MS is clearly written and formatted. The work gives valuable results on spatial extent of NPF events. The article fits well into the topic of the journal. Thus, after considering the following questions and comments as minor revisions, I recommend the publication of the manuscript.

Comments:

[Figure]

1. Page 4., Line 15.: What is the time resolution of the SMPS measurement? It is not included in the text. How were the different systems compared? Were all the instruments running according to the international working standards (Wiedensohler et al., 2012)?

2. Page 5., Line 18.: Title is misleading, since there is nothing about the classification of NPF events. It should be Dynamic parameters of NPF events or similar.

3. Page 5., Line 24.: Were two lognormal modes enough to fit during nucleation? Is there a missing mode?

4. Page 6., Line 6.: Why was the ending height 1500 m at TS? If GDAS does not 'see' the height, then 500 m could be OK as well as it is for the 2 other sites (since it is agl and not msl). If GDAS does 'see' it, then 1500 m agl is too much. Please clarify and reword this.

5. Page 6., Line 12.: One of the crucial questions is the monthly nucleation frequency. There is no data neither in the MS nor the cited paper (Shen et al., 2016b). Do all the 3 sites have the same monthly (annual) nucleation frequency curve? If they do not, it substantially and essentially modifies the whole picture present here. Please provide a paragraph regarding this topic.

6. Page 6., Line 12.: What is the slight difference? Specify and/or explain this with data.

7. Page 18., Fig 4.: On the probability plot: moving from SDZ trough TS to LAN sites, it seems so, that SDZ had N, LAN had NW influence, while TS was a mixed case both from N and NW directions. At the end of Section 3.3 a couple of sentences should be added to highlight (better) the main message of these plots.

Technical requests:

1. Page 2., Line 25.: Sentence should be reworded, in that form it is not well understandable.

2. Page 5., Line 16.: "was be" should be reworded.

3. Table 1.: Missing days should be added with the measurement time interval for specific sites.

4. Figure 6.a: This graph is also explained in the text and it does not provide extra info. I recommend to discard this plot.

---

## Referee Comment (RC2) · Anonymous Referee #2 · 16 Oct 2017

Shen et al. reported the spatial distribution and occurrence probability of regional new particle formation events in eastern China. Regional new particle formation play a profound role in the haze formation over North China Plain. The manuscript is well-written and easy to understand. I suggest that it can be accepted for publication as ACP paper. Here one question was concerned: The high concentration of pre-existing particles may scavenge the condensable vapors and impede the NPF occurrence. As we know, NCP region contains a city cluster. In the atmosphere of urban areas, the particle concentration may be higher than rural or regional background areas. As a result, the NPF may not take place in the urban atmosphere due to a higher condensation sink of condensable vapors. How the heterogeneity of NPF caused by the heterogeneous

pre-existing particle concentrations over a large areas is considered in Nanomap?

---

## Author Comment (AC1) · 31 Oct 2017

The manuscript deals with the spatial extent of new particle formation based on longterm particle number size distribution (PNSD) measurements in China. The work presented here describes the NPF occurrence probability and the possible origin of the nucleating air masses as well. A case study is also performed. The MS is clearly written and formatted. The work gives valuable results on spatial extent of NPF events. The article fits well into the topic of the journal. Thus, after considering the following questions and comments as minor revisions, I recommend the publication of the manuscript. Response: The authors thank the reviewer's comments and try our best

to address the issues point-by point.

Comments: 1. Page 4., Line 15.: What is the time resolution of the SMPS measurement? It is not included in the text. How were the different systems compared? Were all the instruments running according to the international working standards (Wiedensohler et al., 2012)? Response: The raw data derived from SMPS was 5 min resolution. In order to combine with the TDMPS data, 10 min average data was used for the analysis. SMPS system was only operated for half a year at TS, and there was no overlap between SMPS and TDMPS. We could not compare SMPS and TDMPS result directly, but they were operated under the same conditions. The instrument calibration and data inversion routines followed the standards recommended by (Wiedensohler et al., 2012), which assured the data quality and comparability during the long-term measurement at different sites. We also add the information about the SMPS in the text.

2. Page 5., Line 18.: Title is misleading, since there is nothing about the classification of NPF events. It should be Dynamic parameters of NPF events or similar. Response: We totally agreed with the reviewer's comment and revised the tittle to be "Parameters describing NPF events"

3. Page 5., Line 24.: Were two lognormal modes enough to fit during nucleation? Is there a missing mode? Response: Based on our calculation results we found two lognormal modes were enough to parameterize the PNSD on NPF days. On NPF days, the particle concentration usually concentrated in the size below 100 nm. The lognormal fitting algorithm constrained two modes in the size range of 3-25 nm for nucleation mode and 25-100 nm for Aikten mode, respectively, which could capture the nucleation and growth process of NPF event. We didn't consider the accumulation mode (100-1000 nm) as the nucleated particle seldom grew out of 100 nm. We also revised the statement in the text to "In this study, Dp is derived by a fitting algorithm of PNSD based on two lognormal modes (Hussein et al., 2005), which was constrained in the size range of 3-25 nm and 25-100 nm, respectively.".

[Figure]

4. Page 6., Line 6.: Why was the ending height 1500 m at TS? If GDAS does not 'see' the height, then 500 m could be OK as well as it is for the 2 other sites (since it is agl and not msl). If GDAS does 'see' it, then 1500 m agl is too much. Please clarify and reword this. Response: the GDAS data didn't consider the topography of Mt. Tai, with the terrain height is 150 asl. Here we used the ending height of 1500 m agl at TS (equal to 1650 m asl, slightly higher than the evaluation of TS, 1540 m asl) to get rid of the influence by the topography on the air mass calculation. For the other two sites SDZ and LAN, the evaluation was ∼300 m asl and ∼200 m asl, respectively, which was close to the terrain height in the GDAS data and the ending height was chosen to be 500 m was enough. We also revised the sentences in the text to make it clear.

5. Page 6., Line 12.: One of the crucial questions is the monthly nucleation frequency. There is no data neither in the MS nor the cited paper (Shen et al., 2016b). Do all the 3 sites have the same monthly (annual) nucleation frequency curve? If they do not, it substantially and essentially modifies the whole picture present here. Please provide a paragraph regarding this topic. Response: We supplemented a new figure and sentences in the text to describe the occurrence frequency of NPF events observed at three sites. It showed the monthly variation was similar, which was higher in spring and fall, lower in summer. But for SDZ, the NPF also occurred frequently in winter. Fig. 2 was uploaded separately.

6. Page 6., Line 12.: What is the slight difference? Specify and/or explain this with data. Response: The statistical results of NPF parameters based on different length of dataset might be different. For example, the reported NPF frequency at LAN in Shen et al. (2016b) was 28% based on the measurement in 2013. But based on the three-year measurement, the NPF frequency is 22%, which was slightly lower than 28% in 2013. The formation rate and growth rate also showed an annual variation based on the multi year measurement at SDZ, which was also reported in Shen et al. (2016), but was not focused in this study. We revised the sentences in the text and made the statement more clear.
7. Page 18., Fig 4.: On the probability plot: moving from SDZ trough TS to LAN sites, it seems so, that SDZ had N, LAN had NW influence, while TS was a mixed case both from N and NW directions. At the end of Section 3.3 a couple of sentences should be added to highlight (better) the main message of these plots. Response: we added few sentences in the last paragraph in section 3.3 to highlight the ideas as the reviewer suggested. "It was also found that the high NPF occurrence probability was under the conditions of northerly air masses at SDZ, northwesterly and southerly air masses at LAN. While at TS, the probability was influenced by the mixing of northerly and southeasterly air mass, although high uncertainty exited due to the short length of dataset."

Technical requests: 1. Page 2., Line 25.: Sentence should be reworded, in that form it is not well understandable. Response: The sentence was revised to be " In China, the particle pollution is a serious issue and there have been several studies addressing the importance of NPF events based on long-term field measurement in recent years (Shen et al., 2011...)".

2. Page 5., Line 16.: "was be" should be reworded. Response: in the text, "be" was moved.

3. Table 1.: Missing days should be added with the measurement time interval for specific sites. Response: the missing days with time interval has been added in Table 1.

4. Figure 6.a: This graph is also explained in the text and it does not provide extra info. I recommend to discard this plot. Response: fig. 6a has been removed

[Figure]

[Figure]

**Fig. 1.**

---

## Author Comment (AC2) · 31 Oct 2017

Shen et al. reported the spatial distribution and occurrence probability of regional new particle formation events in eastern China. Regional new particle formation play a profound role in the haze formation over North China Plain. The manuscript is well-written and easy to understand. I suggest that it can be accepted for publication as ACP paper. Here one question was concerned: The high concentration of pre-existing particles may scavenge the condensable vapors and impede the NPF occurrence. As we know, NCP region contains a city cluster. In the atmosphere of urban areas, the particle concentration may be higher than rural or regional background areas. As a result,

the NPF may not take place in the urban atmosphere due to a higher condensation sink of condensable vapors. How the heterogeneity of NPF caused by the heterogeneous pre-existing particle concentrations over a large areas is considered in Nanomap? Response: Thanks very much for reviewer's comments. High concentration of pre-existing particles (high condensation sink) is not favorable for NPF events due to the competition between scavenging process and nucleation process of the precursors. Although the particle number concentration can be higher in urban site as expected, there have been several studies reporting high NPF frequency in urban sites. For example, Wu et al (2007) has reported that NPF frequency in Beijing was about 40%, which was close to the frequency of 36% at SDZ reported by Shen et al (2011). The pre-existing particle concentration was influenced by the air mass origin. NPF event showed higher frequency when clean and dryer air mass from northwest was dominated in Beijing and lower when polluted air mass from south. Wang et al. (2013) reported that the NPF event occurred simultaneously at rural site SDZ and urban site PKU, which located about 100 km apart and accounted for ∼70% of the total NPF events at PKU site. The result showed the regional NPF event could be homogenous at least 100 km under the influence by the northwesterly air masses from Mongolia and with the synoptic of high-pressure system. Furthermore, the regional NPF events were also observed at TS and SDZ (500 km apart) simultaneously under the influence of homogenous surface high-pressure system and discussed in the text. It can be concluded that the homogeneity of regional NPF event related with the synoptic conditions. It has been clarified in the text that only regional NPF events, which can extend to hundreds of kilometers, were discussed by NanoMap method in the study. But it's true that we can't make sure that each regional NPF event is homogenies. Especially the start and end time of NPF event in a large regional scale would be heterogeneous and the uncertainty is hard to evaluate. In the section 2.3, we have clarified that we made assumptions when NanoMap is applied and we assumed the formation of particles is assumed to take place at the same time in a large area and is starting at the same time throughout the region. These uncertainties may mask local differences of NPF event in each geographical position. But the long dataset can help to reduce the uncertainties and we have clarified that in the text.

---

## Author Response (AR2)

The manuscript deals with the spatial extent of new particle formation based on longterm particle number size distribution (PNSD) measurements in China. The work presented here describes the NPF occurrence probability and the possible origin of the nucleating air masses as well. A case study is also performed. The MS is clearly written and formatted. The work gives valuable results on spatial extent of NPF events. The article fits well into the topic of the journal. Thus, after considering the following questions and comments as minor revisions, I recommend the publication of the manuscript.

**Response:** The authors thank the reviewer's comments and try our best to address the issues point-by point.

Comments:

1. Page 4., Line 15.: What is the time resolution of the SMPS measurement? It is not included in the text. How were the different systems compared? Were all the instruments running according to the international working standards (Wiedensohler et al., 2012)?
   **Response:** The raw data derived from SMPS was 5 min resolution. In order to combine with the TDMPS data, 10 min average data was used for the analysis. SMPS system was only operated for half a year at TS, and there was no overlap between SMPS and TDMPS. We could not compare SMPS and TDMPS result directly, but they were operated under the same conditions. The instrument calibration and data inversion routines followed the standards recommended by (Wiedensohler et al., 2012), which assured the data quality and comparability during the long-term measurement at different sites. We also add the information about the SMPS in the text in P 4 L13-17.

2. Page 5., Line 18.: Title is misleading, since there is nothing about the classification of NPF events. It should be Dynamic parameters of NPF events or similar.
   **Response:** We totally agreed with the reviewer's comment and revised the tittle to be "Parameters describing NPF events" in P5L20.

3. Page 5., Line 24.: Were two lognormal modes enough to fit during nucleation? Is there a missing mode?
   **Response:** Based on our calculation results we found two lognormal modes were enough to parameterize the PNSD on NPF days. On NPF days, the particle concentration usually concentrated in the size below 100 nm. The lognormal fitting algorithm constrained two modes in the size range of 3-25 nm for nucleation mode and 25-100 nm for Aikten mode, respectively, which could capture the nucleation and growth process of NPF event. We didn't consider the accumulation mode (100-1000 nm) as the nucleated particle seldom grew out of 100 nm. We also revised the statement in the text to "In this study, $D_p$ is derived by a fitting algorithm of PNSD based on two lognormal modes (Hussein et al., 2005), which was constrained in the size range of 3-25 nm and 25-100 nm, respectively." In P5L18.

4. Page 6., Line 6.: Why was the ending height 1500 m at TS? If GDAS does not 'see' the height, then 500 m could be OK as well as it is for the 2 other sites (since it is agl and not msl). If GDAS does 'see' it, then 1500 m agl is too much. Please clarify and reword this.
**Response:** the GDAS data didn't consider the topography of Mt. Tai, with the terrain height is 150 asl. Here we used the ending height of 1500 m agl at TS (equal to 1650 m asl, slightly higher than the evaluation of TS, 1540 m asl) to get rid of the influence by the topography on the air mass calculation. For the other two sites SDZ and LAN, the evaluation was ~300 m asl and ~200 m asl, respectively, which was close to the terrain height in the GDAS data and the ending height was chosen to be 500 m was enough. We revised the sentences in the text to make it clear in P6L5-9.

5. Page 6., Line 12.: One of the crucial questions is the monthly nucleation frequency. There is no data neither in the MS nor the cited paper (Shen et al., 2016b). Do all the 3 sites have the same monthly (annual) nucleation frequency curve? If they do not, it substantially and essentially modifies the whole picture present here. Please provide a paragraph regarding this topic.
**Response:** We supplemented a new figure and sentences in the text to describe the occurrence frequency of NPF events observed at three sites. It showed the monthly variation was similar, which was higher in spring and fall, lower in summer. But for SDZ, the NPF also occurred frequently in winter. We also revised the sentences in P6L16-18, L25-29.

[Figure]

Fig. 2. The monthly average frequency of NPF event occurrence at SDZ, TS and LAN, respectively.

6. Page 6., Line 12.: What is the slight difference? Specify and/or explain this with data.
**Response:** The statistical results of NPF parameters based on different length of dataset might be different. For example, the reported NPF frequency at LAN in Shen et al. (2016b) was 28% based on the measurement in 2013. But based on the three-year measurement, the NPF frequency is 22%, which was slightly lower than 28% in 2013. The formation rate and growth rate also showed an annual variation based on the multi year measurement at SDZ, which was also reported in Shen et al. (2016), but was not focused in this study. We revised the sentences in the text and made the statement more clear in P6L16-18, L25-29.

7. Page 18., Fig 4.: On the probability plot: moving from SDZ trough TS to LAN sites, it seems so, that SDZ had N, LAN had NW influence, while TS was a mixed case both from N and NW directions. At the end of Section 3.3 a couple of sentences should be added to highlight (better) the main message of these plots.
**Response:** we added few sentences in the last paragraph in section 3.3 to highlight the ideas as the reviewer suggested. "It was also found that the high NPF occurrence probability was under the conditions of northerly air masses at SDZ, northwesterly and southerly air masses at LAN. While at TS, the probability was influenced by the northerly and southeasterly air mass, respectively, although high uncertainty exited due to the short length of dataset." in P8L21-24.

Technical requests:
1. Page 2., Line 25.: Sentence should be reworded, in that form it is not well understandable.
**Response:** The sentence was revised to be " In China, the particle pollution is a serious issue and there have been several studies addressing the importance of NPF events based on long-term field measurement in recent years (Shen et al., 2011…)" in P2L25.

2. Page 5., Line 16.: "was be" should be reworded.
**Response:** in the text, "be" was moved.

3. Table 1.: Missing days should be added with the measurement time interval for specific sites.
**Response:** the missing days with time interval has been added in Table 1.

4. Figure 6.a: This graph is also explained in the text and it does not provide extra info. I recommend to discard this plot.
**Response:** fig. 6a has been removed

**Anonymous Referee #2**

Shen et al. reported the spatial distribution and occurrence probability of regional new particle formation events in eastern China. Regional new particle formation play a profound role in the haze formation over North China Plain. The manuscript is well-written and easy to understand. I suggest that it can be accepted for publication as ACP paper. Here one question was concerned: The high concentration of pre-existing particles may scavenge the condensable vapors and impede the NPF occurrence. As we know, NCP region contains a city cluster. In the atmosphere of urban areas, the particle concentration may be higher than rural or regional background areas. As a result, the NPF may not take place in the urban atmosphere due to a higher condensation sink of condensable vapors. How the heterogeneity of NPF caused by the heterogeneous pre-existing particle concentrations over a large areas is considered in Nanomap?

**Response**: Thanks very much for reviewer's comments. High concentration of pre-existing particles (high condensation sink) is not favorable for NPF events due to the competition between scavenging process and nucleation process of the precursors. Although the particle number concentration can be higher in urban site as expected, there have been several studies reporting high NPF frequency in urban sites. For example, Wu et al (2007) has reported that NPF frequency in Beijing was about 40%, which was close to the frequency of 36% at SDZ reported by Shen et al (2011). The pre-existing particle concentration was influenced by the air mass origin. NPF event showed higher frequency when clean and dryer air mass from northwest was dominated in Beijing and lower when polluted air mass from south. Wang et al. (2013) reported that the NPF event occurred simultaneously at rural site SDZ and urban site PKU, which located about 100 km apart and accounted for ~70% of the total NPF events at PKU site. The result showed the regional NPF event could be homogenous at least 100 km under the influence by the northwesterly air masses from Mongolia and with the synoptic of high-pressure system. Furthermore, the regional NPF events were also observed at TS and SDZ (500 km apart) simultaneously under the influence of homogenous surface high-pressure system and discussed in the text. It can be concluded that the homogeneity of regional NPF event related with the synoptic conditions.

It has been clarified in the text that only regional NPF events, which can extend to hundreds of kilometers, were discussed by NanoMap method in the study. But it's true that we can't make sure that each regional NPF event is homogenies. Especially the start and end time of NPF event in a large regional scale would be heterogeneous and the uncertainty is hard to evaluate. In the section 2.3, we have clarified that we made assumptions when NanoMap is applied and we assumed the formation of particles is

assumed to take place at the same time in a large area and is starting at the same time throughout the region. These uncertainties may mask local differences of NPF event in each geographical position. But the long dataset can help to reduce the uncertainties and we have clarified that in the text.

Reply to reviewer 3

We appreciate the constructive suggestions made by the third reviewer and editor, which have improved the revision of our original manuscript. Our responses to the reviewer comments are given in black font below and the comments by reviewer were highlighted by green.

Comments:

1. Page 2, line 5: The authors cited previous study (Guo et al., 2014): "... and contribute to the particle mass enhancement even in polluted regions." Is this conclusion/statement correct? In my opinion, although many NPF events were regularly followed by an increasing of particle mass, this only indicates a high abundance of condensable vapors in the atmosphere. Actually, the newly formed particles can only grow up to 60~100 nm. The contribution by particles smaller than 100 nm to aerosol mass is not significant. Maybe the large number of grown particles can, through the coagulation process, grow into a larger size range, where they could contribute more efficiently to the particle mass concentration or particle extinction. But I think it is not proper, or at least it may result in certain misunderstanding, to directly connect the NPF (or continuous growth of nucleated particles) with haze formation. Please consider this issue.

Response: I agree with the reviewer's comments that the nucleated particles usually grow up to the sizes of 60-100 nm, which is not considerable to the mass concentration. But in the study by Guo et al. (2014), the typical NPF events governed by meteorological conditions were reported. During the two NPF cases, the contribution from primary emissions and regional transport is minor. NPF produce large number concentration of small particle, which can grow continuously from nucleation mode over multiples days to yield numerous larger particles. In the polluted conditions or the air mass suddenly changed, the nucleated mode particle perhaps can't grow further, but for the cases reported by Guo et al, the growth process did last for several days. So we revised the sentences in the text and added the conditions to constrain the cited conclusion. It was changed to "and contribute to the particle mass enhancement even in polluted regions for some NPF cases governed by meteorological conditions with minor contribution from primary emissions and regional transport (Guo et al., 2014)." in P2L5.

2. Page 2, lines 6-7: What is "possible particle source"? Do you mean where it come from? Please explain.

Response: "possible particle source" means where the nucleation mode particle possibly come from. The sentence has been changed to "where the nucleation mode particles possibly come from".

3. Page 2, lines 20-21: Any explanation? Is it true for the NPF event in polluted area?

Response: The study by Hussein et al. (2009) was conducted in Finland and Sweden, which are quite clean region. The NPF events can usually last for one or two days, which is different from polluted urban areas, like Beijing, where NPF events are usually interrupted by high concentration of pre-existing particles. But in this work, we focused on three rural sites, which is relative clean as compared with the polluted area in China. Furthermore, only regional NPF events were analyzed, which usually occur under clean circumstance. We

revised this sentence to "The study conducted by Hussein et al (2009) at the clean region of Finland and Sweden showed that" to make it clear.

4. Page 2, lines 25-27: Please cite the recent two overview papers: Kulmala et al., (2016) and Wang et al., (2017).

Response: the two references suggested by the reviewer have been added.

5. Page 4, line 15: 3080 is the type of the classifier, not SMPS. Should be 3936, I guess. Please check.

Response: yes, the reviewer is right. The DMA model is 3080 and SMPS model is 3936. We have changed in the text.

6. Page 5, line 14: How you consider the size dependence of GR. What is the uncertainty?

Response: the growth rate during the nucleation growth process is not linear and also dependents on the selected size range in the calculation. In this work, we calculated GR as the change rate of geometric mean diameter of PNSD from the NPF start time to the end of growth. In Nanomap method, the GR is used to calculate the time-shift between the observed formation time (burst of 3 nm) and the real formation time (1 nm). But the growth rate from 1-3 nm can't be derived in this work. This will underestimate the time-shift as GR in the smaller size range should be lower. It will cause some uncertainty in the nucleating air mass calculation depends on time point we determined for nucleation start and end time, as well as the end time of growth. But based on the long length of dataset, the uncertainty should be reduced.

7. For the section 2.3 Nanomap, I would suggest that several issues should be explained more clearly:

1. Based on the Kristensson et al., 2014, it seems that the maximum distance is 500 km in this method. I assume this value is determined by the air mass trajectory calculated for Finland case. So could we directly use this value for other studies? Have you compared the air mass trajectory patterns between Finland and China? Also, as one reviewer mentioned, 500 km in China contains a lot of megacities with heavily emissions, so the first assumption in Nanomap should be reconsidered.

2. It seems that this method is also closely related to the calculation of air mass trajectory, so why you use 72 h, have you tried others (48- and 96-h), any differences? In Section 3.2, you want to retrieve where does the NPF take place, but I think it also depends on that how you calculate the air mass trajectory.

Response: (1) The study based on 7-year dataset at SDZ showed quite similar result as that in Hyytiälä conducted by Kristensson et al (2014), which NPF event occurred frequently under the influence by the typical air masses or synoptic condition. At SDZ, the northwesterly air masses were dominated and the trajectory count and NPF occurrence probability result (Fig. 4 in the text) had proved that the back trajectories were constrained within the distance of 500 km. But at LAN station in YRD region, the maximum horizontal extension of the observed NPF event according to NanoMap is restricted to within about 200 km due to the low wind speed as we had illustrated in the text. At TS station, most back trajectories were also in the range of 500 km, but the NPF occurrence probability had large uncertainty due to the complex topography and short length of dataset, which was also clarified in the text.

The first assumption in this method has been discussed by Kristensson et al. (2014) and is not valid in every situation. It weakens the potential local differences in the location of

formation events. But fortunately, we found NPF event occurred simultaneously at SDZ and TS, which located about 500 km. This proved that regional NPF event could occur over large spatial extent influenced by the large scale synoptic condition.

(2) NanoMap method has been successfully applied in other studies and was validated to estimate the NPF occurrence with 500 km. From Fig. 4 in the text, we can find most trajectories were within the distance of 500 km. 48h-trajectories calculation result can't reach the distance of 500 km and 96h-trajectories will result more uncertainties, as we have shown that most air mass at LAN were regional that actually constrained to the distance of 200 km. In optimization, we chose 72-h back trajectories for analysis.

8. Section 3.3: How you calculate the occurrence probability? Please classify.

Response: in section 2.3 we has introduced NanoMap method and it has been clarified that "
[revised manuscript text omitted]